# Probiotic powder ameliorates colorectal cancer by regulating *Bifidobacterium animalis, Clostridium cocleatum,* and immune cell composition

Xiaojuan Yang[1]☯, Qian Cao[2]☯, Bin Ma[3]☯, Yuhan Xia[4], Miao Liu[2], Jinhua Tian[2], Jian Chen[5], Chunxia Su[6]*, Xiangguo Duan [2,7]*

1 School of Basic Medical Sciences, Ningxia Medical University, Yinchuan, China, 2 College of Clinical Medicine, Ningxia Medical University, Yinchuan, China, 3 Department of Oncology Surgery, The First People's Hospital of Yinchuan, Yinchuan, China, 4 Department of Nutrition, General Hospital of Ningxia Medical University, Yinchuan, China, 5 Guolong Hospital, Yinchuan, China, 6 Department of Pathogen Biology and Immunology, School of Basic Medical Science, Ningxia Medical University, Yinchuan, China, 7 Department of Laboratory Surgery, General Hospital of Ningxia Medical University, Yinchuan, China

☯ These authors contributed equally to this work.
* 1651085195@qq.com (SC); 2455281549@qq.com, duanxiangguo@nxmu.edu.cn (DX)

**Data Availability Statement:** All relevant data are within the paper and its Supporting Information files.

## Abstract

Based on the relationship between the gut microbiota and colorectal cancer, we developed a new probiotic powder for treatment of colorectal cancer. Initially, we evaluated the effect of the probiotic powder on CRC using hematoxylin and eosin staining, and evaluated mouse survival rate and tumor size. We then investigated the effects of the probiotic powder on the gut microbiota, immune cells, and apoptotic proteins using 16S rDNA sequencing, flow cytometry, and western blot, respectively. The results showed that the probiotic powder improved the intestinal barrier integrity, survival rate, and reduced tumor size in CRC mice. This effect was associated with changes in the gut microbiota. Specifically, the probiotic powder increased the abundance of *Bifidobacterium animalis* and reduced the abundance of *Clostridium cocleatum*. In addition, the probiotic powder resulted in decreased numbers of CD4$^+$ Foxp3$^+$ Treg cells, increased numbers of IFN-γ$^+$ CD8$^+$ T cells and CD4$^+$ IL-4$^+$ Th2 cells, decreased expression of the TIGIT in CD4$^+$ IL-4$^+$ Th2 cells, and increased numbers of CD19$^+$ GL-7$^+$ B cells. Furthermore, the expression of the pro-apoptotic protein BAX was significantly increased in tumor tissues in response to the probiotic powder. In summary, the probiotic powder ameliorated CRC by regulating the gut microbiota, reducing Treg cell abundance, promoting the number of IFN-γ$^+$ CD8$^+$ T cells, increasing Th2 cell abundance, inhibiting the expression of TIGIT in Th2 cells, and increasing B cell abundance in the immune microenvironment of CRC, thereby increasing the expression of BAX in CRC.

**Funding:** This work was supported by National Natural Science Foundation of China (No. 81760437,82160548),Natural Science Foundation of Ningxia (2020AAC03183, 2021AAC02011).

**Competing interests:** The authors declare no potential conflicts of interest with respect to the research, authorship, and/or publication of this article.

## Introduction

Colorectal cancer (CRC) is a leading cause of cancer-related mortality [1]. Despite considerable developments such as advanced surgical procedures and chemotherapeutic drugs, CRC remains a leading causes of cancer-related death [2]. Therefore, prevention and treatment of CRC remains a challenge.

Recent studies have focused on the impact of the gut microbiota in CRC [3]. Studies have shown that gut microbiota imbalances are closely related to the occurrence of CRC. Specifically, increases in *Bacteroides*, *Fusobacterium*, and *Campylobacter*, and decreases in beneficial bacteria (e.g, *Bifidobacterium* and *Lactobacillus*) have been observed in patients with CRC [4]. Therefore, regulation of the gut microbiota may play an important role in achieving CRC remission.

Studies have shown that probiotic-based preparations can prevent gut microbiota disorders, and probiotics can be used as an effective therapy against CRC [5]. We previously showed that decreased abundance of lactic acid bacteria was closely related to CRC. Lactic acid bacteria are the most common types of microbes used as probiotics, and can induce CRC remission through modulation of the gut microbiota composition [6]. *Bifidobacterium* and *Lactobacillus* are the most commonly used lactic acid bacteria. In addition, prebiotics (e.g., stachyose, isomalto-oligosaccharide, and the newly discovered *Lycium barbarum* polysaccharide) can promote growth of beneficial bacteria, resulting in augmentation of the effects of probiotics [7]. We developed a new formula of probiotic powder to improve understanding of, and treatment strategies for, CRC

A growing body of evidence indicates that the gut microbiota is critical to intestinal immunity, and plays an important role in activation of T cells and B cells [8]. We evaluated whether regulation of the gut microbiota using probiotics could alter the immune microenvironment in CRC mice. To better understand changes in the immune microenvironment, we evaluated changes in the immune cell inhibitory receptor, T-cell immunoreceptor with immunoglobulin and tyrosine–based inhibitory motif domain (TIGIT). T-cell immunoreceptor with immunoglobulin and tyrosine–based inhibitory motif domain belongs to the family of poliovirus receptors (PVRs) that competes with CD226, which is the co-stimulatory receptor of T cells and NK cells, by binding to CD112 and CD155 (PVR) expressed on antigen-presenting cells (APCs) and tumor cells. A previous study reported that patients with CRC had relatively higher expression of TIGIT in T cells [9]. We investigated the effects of a probiotic powder on B cells, T cells, and TIGIT through regulation of the gut microbiota.

## Materials and methods

### Materials

Azoxymethane (CAS No.:25843-45-2, molecular weight: 74.08) and dextran sulfate sodium salt (CAS No.:9011-18-1, molecular weight: ~40,000) were purchased from Sigma Aldrich (Shanghai) Trading Co., Ltd (Shanghai, China). *Bifidobacterium* adolescence powder ($5.0 \times 10^{10}$ CFU/g) and *Lactobacillus* plantarum powder ($3.0 \times 10^{11}$ CFU/g) were purchased from Jiangsu Wecare Biotechnology Co., Ltd. (Jiangsu, China). Stachyose, isomalto-oligosaccharide, and maltodextrin were purchased from Shanghai Yuanye Bio-Technology Co., Ltd. (Shanghai, China). *Lycium barbarum* polysaccharides were purchased from Bairuiyuan Gouqi Co., Ltd. (Yinchuan, China). Specific-pathogen-free (SPF)-grade AIN-93G feed was purchased from XiaoShu YouTai Biotechnology Co., Ltd. (Beijing, China).

## Preparation of probiotic powder

Table 1 shows the composition of the probiotic powder. All ingredients of the probiotic powder were food-grade materials. According to the corresponding proportions, the ingredients were fully mixed in a three-dimensional mixer, and the mixed probiotic powder was stored at -20°C. Prior to use, the powder was fully dissolved in physiological saline.

## Mice

Specific pathogen-free C57BL/6J male mice (age: 6 weeks; weight: 18–22 g) were purchased from Beijing HuaFukang Biotechnology Co., Ltd. (Beijing, China). All mice were housed in an SPF animal facility in the Ningxia Medical University Laboratory Animal center at an ambient temperature of approximately 22°C under a 12 hour light-dark cycle. The mice were allowed free access to food and water. Animal experiments were performed in compliance with the Guide for the Care and Use of Laboratory Animals and approved by the Ethics Committee of Ningxia Medical University (Yinchuan, China, approval number: IACUC-NYLAC-2019-023). Following the experiments, the mice were sacrificed by cervical vertebra dislocation. This study was performed according to the international, national, and institutional rules for animal experiments, clinical studies, and biodiversity rights.

## Feed

C57BL/6J mice were divided into three groups with 18 mice in each group as follows: normal mouse control group (NC); CRC mouse control group (RC); and CRC mouse probiotic powder intervention group (CP). The mouse CRC model was induced using the azoxymethane/ dextran sulfate sodium salt (AOM/DSS) method, as shown in the figure below. All mice were fed SPF-grade AIN-93G feed. Mice in the CP group received the probiotic solution through intragastric administration. The preparation administered to each mouse consisted of 0.3 mL of physiological saline containing 6.6 mg of probiotic powder, once daily (Fig 1).

## Hematoxylin and eosin staining

Colorectal tissue was collected, fixed in 4% neutral-buffered paraformaldehyde, embedded in paraffin, and processed for histological analysis. Tissues sections were subjected to hematoxylin and eosin staining, and tissue changes were observed under a microscope.

## Isolation of mononuclear cells

Spleens were removed and homogenized in 2 mL of phosphate-buffered saline (PBS). The homogenates were filtered through a 300-mesh filter, and rinsed with 2 mL of PBS to obtain a

**Table 1. Composition of the probiotic powder.**

| Name | Content (/g) |
|---|---|
| *Lactobacillus plantarum* powder | 18%-24% |
| *Bifidobacterium* adolescence powder | 18%-24% |
| Stachyose | 20%-23% |
| Lycium barbarum polysaccharides | 10%-16% |
| Isomalto-oligosaccharide | 2%-5% |
| Maltodextrin | Margin[a] |

[a] The margin is equal to 100% minus the content of the other ingredients.

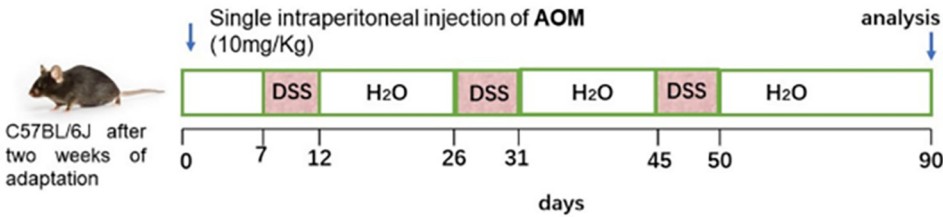

**Fig 1. The mouse CRC model was induced using the azoxymethane/dextran sulfate sodium salt method.**

single-cell suspension. Mouse lymphocyte separation solution (2 mL) was added, and the mixture was centrifuged for 20 min at 4°C at 2,000 rpm. Lymphocytes were collected, PBS was added, and the mixture was centrifuged to obtain mononuclear cells.

## Flow cytometry

A BD FACSCelesta™ flow cytometer was used to detect cells. The antibodies used in the experiments included AF488-conjugated anti-CD4 antibody (BD Biosciences), FITC-conjugated anti-CD19 antibody (BD Biosciences), BV510-conjugated anti-CXCR3 antibody (BD Biosciences) and phycoerythrin-conjugated anti-TIGIT antibody (BD Biosciences), PE-CF594-conjugated anti-CD8 antibody (BD Biosciences). These were surface antibodies, and were detected after staining in the dark for 30 min. Phycoerythrin-conjugated anti-forkhead box P3 (anti-Foxp3) antibody (BD Biosciences), BV421-conjugated anti-interleukin 4 (anti-IL-4) antibody (BD Biosciences), PerCP-Cy™5.5 -conjugated RoRrt antibody (BD Biosciences), APC-Cy7-conjugated anti-IFN-γ antibody (BD Biosciences) and AF647-conjugated anti-GL-7 antibody (BD Biosciences) are intracellular antibodies, and the cell membrane needed to be disrupted for 40 min prior to staining.

## Fecal collection

A warm and damp cotton swab was used to stimulate the anuses of the mice. The collected feces were placed into a sterile cryotube and immediately submerged in liquid nitrogen. The following day, the feces were transferred to a -80°C freezer for storage.

## 16S rDNA sequencing

Sequencing of the gut microbiota was performed by NovoGene Company (China, Beijing), and the data were analyzed using the NovoMagic platform. The steps included the following [10–12]:

## Extraction of genomic DNA

Total genomic DNA from samples was extracted using the CTAB/SDS method. The DNA concentration and purity were evaluated using 1% agarose gels. 16S rRNA/18SrRNA/ITS genes of distinct regions were amplified using specific primers with barcodes. All PCR reactions were performed using 15 μL of Phusion® High-Fidelity PCR Master Mix (New England Biolabs) containing 0.2 μM forward and reverse primers, and about 10 ng of template DNA. Thermal cycling consisted of initial denaturation at 98°C for 1 min, followed by 30 cycles of denaturation at 98°C for 10 s, annealing at 50°C for 30 s, elongation at 72°C for 30 s, and a hold at 72°C for 5 min.

## PCR products quantification and qualification

Equal volumes of 1X loading buffer (containing SYBR green) and PCR products were mixed and separated using a 2% agarose gel. The PCR products was mixed in equal densities, then purified using a Qiagen Gel Extraction Kit (Qiagen, Germany).

## Library preparation and sequencing

Sequencing libraries were generated using TruSeq® DNA PCR-Free Sample Preparation Kit (Illumina, USA) following manufacturer's instructions, and index codes were added. Library quality was assessed using a Qubit@ 2.0 Fluorometer (Thermo Scientific) and Agilent Bioanalyzer 2100 system. The library was sequenced on an Illumina NovaSeq platform and 250 bp paired-end reads were generated.

## Western blot analysis

A protein extraction kit (KeyGen Biotech Co. Ltd., Nanjing, China) was used to extract protein from tumor tissue, and the protein concentration was determined using a BCA protein quantitation kit (KeyGen Biotech Co. Ltd., Nanjing, China). Thirty micrograms of protein were separated by 10% SDS PAGE, then transferred to PVDF membranes (Millipore, USA). After blocking with 5 percent skim milk, the membranes were incubated with antibodies against GAPDH (1:2,000 cat. no.TA-08; BIOSS, Beijing, China), BAX (1:1,000 cat. no.60267-1-Ig proteintech), and Bcl-2 (1:1,500 cat. no.26593-1-AP proteintech) antibodies at 4°C overnight. Then, the membranes were incubated with goat anti-mouse or rabbit IgG horseradish peroxidase (HRP)-conjugated secondary antibodies (1:5,000; cat. nos. A21010 and A21020; Abbkine Scientific Co., Ltd.) for 1 h at room temperature, and signals were detected by ECL chemiluminescence and measured using ImageJ software.

## Statistical analysis

Continuous variables are expressed as the median and the range. Statistical comparisons were performed using analysis of variance. Data are expressed as the mean ± standard deviation. A two-sided P-value <0.05 denoted statistically significant differences.

# Results

## Effects of the probiotic powder

We evaluated intestinal tissue, tumor size, and survival of mice following intervention with the probiotic powder to assess whether this treatment could ameliorate CRC. In mouse colorectal tissue obtained from the NC group, we observed intact finger-like elongated structures and compact, neatly arrayed epithelium. However, CRC caused severe breakage and incomplete tissue, accompanied by a decrease in goblet cells. Interestingly, compared with the RC group, which exhibited more inflammatory necrosis and crypt abscess, the CP group had more goblet cells, inflammatory cell infiltration, and a relatively complete tissue structure (Fig 2A–2C). In addition, compared with the RC group, the survival rate of mice in the CP group was significantly improved and the tumor volume was decreased (Fig 2D–2E). Tumor formation rates, tumor size, and time to death are shown in Tables 2 and 3. Our data showed that treatment with the probiotic powder ameliorated CRC.

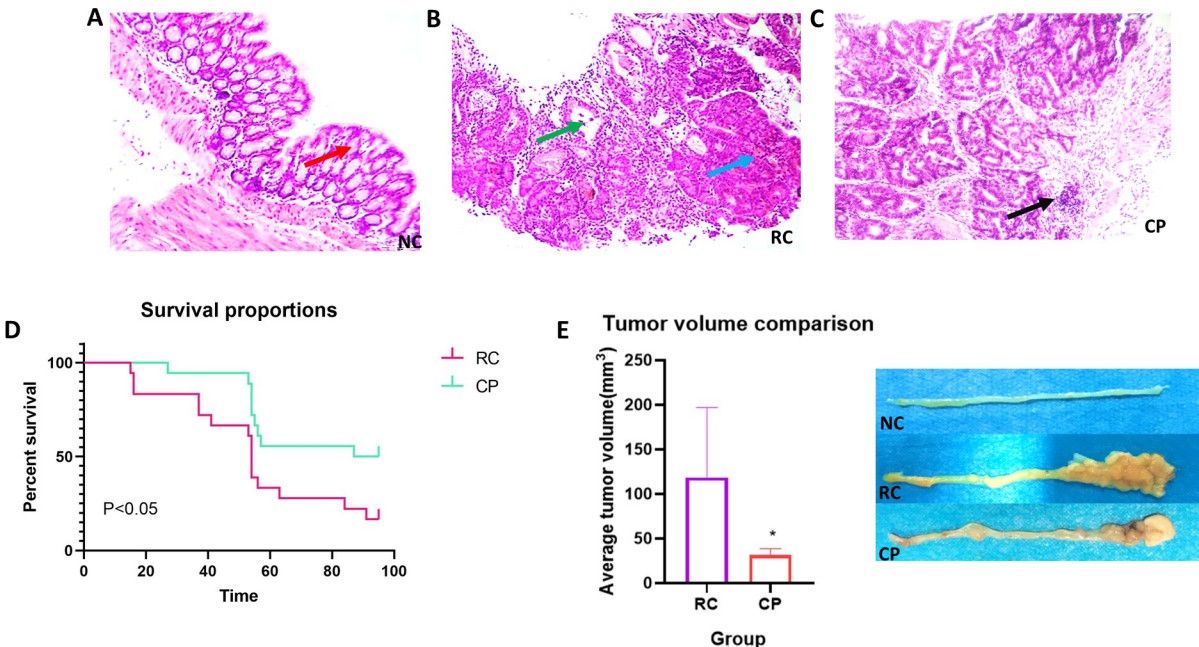

**Fig 2. Effect of the probiotic powder on colorectal cancer. (A, B, C)** Hematoxylin and eosin (HE) staining of the NC, RC, and CP groups, respectively (×400). Mouse colorectal tissue was obtained at the 13th week, and the histological changes of mouse colorectal tissues were analyzed using HE. The arrows represent important tissue damage. The red arrow represents normal goblet cells, the green arrow represents crypt abscess, the blue arrow represents inflammatory necrosis, and the black arrow represents inflammatory cell infiltration. (**D**) Survival rate of mice. The death of mice was recorded every day from initiation of the mouse model, and survival curves were generated through the 90th day. (**E**) Tumor volume of mice. The minimum and maximum diameters of tumors on each colorectal tissue sample were measured using Vernier calipers, and the volume of each tumor and the average tumor volume of each mouse were calculated. The tumor volume was $0.5 \times$ the minimum diameter of the tumor $\times$ the square of the maximum diameter of the tumor. One way analysis of variance (ANOVA), *$p < 0.05$.*

## Probiotic powder improved gut microbiota dysbiosis in CRC

Studies have shown that the gut microbiota is closely associated with colorectal cancer. Therefore, we investigated whether the effect of the probiotic powder on CRC was related to changes in the gut microbiota [13]. First, we analyzed the effect of the probiotic powder on the composition of the gut microbiota. The results of the principal coordinates analysis (PCoA) showed that there were significant differences between the three groups. However, extension of the intervention time reduced the difference between the CP and NC groups (Fig 3A). The Venn diagrams in Fig 3B show that the common and unique operational taxonomic units among the NC, RC, and CP groups differed at different time points. At any given time point, there were more operational taxonomic units in common between the NC and CP groups than between the NC and RC groups (Fig 3B). These results showed that the gut microbiota was imbalanced

**Table 2. The occurrence of tumors in mice.**

| Group | Tumor formation rate | Average tumor volume(mm³)[a] | | | | |
|---|---|---|---|---|---|---|
| | | Mice 1 | Mice 2 | Mice 3 | Mice 4 | Mice 5 |
| RC | 100% | 81.39 | 66.06 | 208.73 | | |
| CP | 100% | 29.23 | 43.23 | 24.12 | 32.49 | 28.43 |

[a] Average tumor volume: The average tumor volume is the average of all tumor volumes in each mouse, and the tumor volume is $0.5 \times$ the minimum diameter of the tumor $\times$ the square of the maximum diameter of the tumor.

**Table 3. Death of mice in each group.**

| Group | Day 10 | | | Day 30 | | | Day 60 | | | Day 90 | | |
|---|---|---|---|---|---|---|---|---|---|---|---|---|
| | N[a] | ND[b] | NS[c] | N | ND | NS | N | ND | NS | N | ND | NS |
| RC | 18 | 0 | 18 | 18 | 3 | 15 | 15 | 9 | 6 | 6 | 2 | 4 |
| CP | 18 | 0 | 18 | 18 | 1 | 17 | 17 | 7 | 10 | 10 | 1 | 9 |

[a] N is the total number of mice in a group over a certain period of time

[b] ND is the number of dead mice

[C] NS is the number of survivors

in CRC, and administration of probiotic powder could improve gut microbiota dysbiosis in CRC.

## Key bacteria affected by the probiotic powder in CRC

Next, we focused on determining the bacteria affected by treatment with the probiotic powder at different time points using MetStat analysis. Fig 4A and 4B show a histogram of species relative abundance at the level of phylum and genus, respectively. As shown in Fig 4C–4F, at week

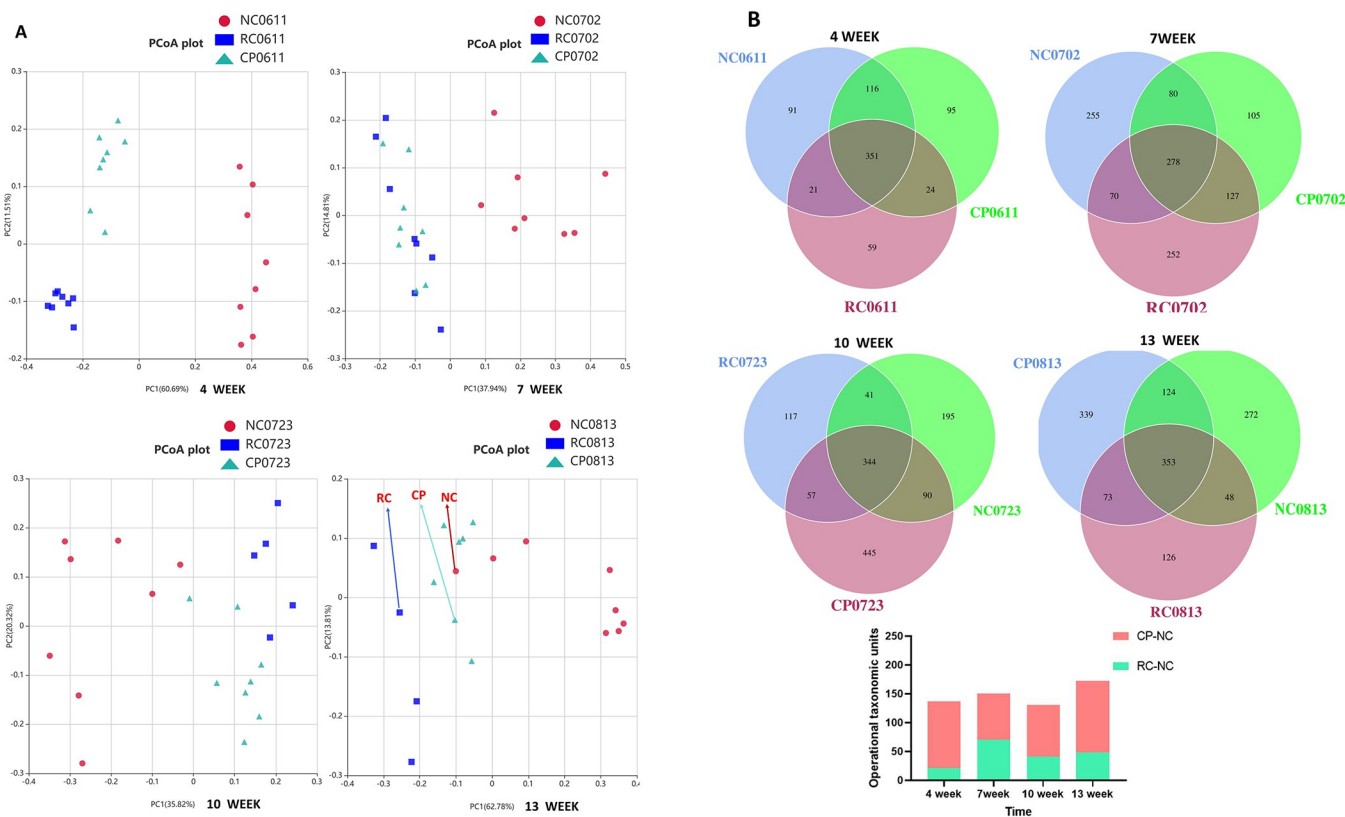

**Fig 3. Effect of the probiotic powder on the gut microbiota structure.** (**A, B**) PCoA analysis and Venn diagrams of mice at weeks 4, 7, 10, and 13, respectively. PCoA analysis (Principal Co-ordinates Analysis) was used to extract the most important elements and structures from multi-dimensional data by sorting a series of eigenvalues and eigenvectors, and then to select the combination of principal coordinates with the greatest contributions. Closer distances indicate more similarity in species composition. Therefore, samples with highly similar community structures tended to cluster more closely, and samples that differed in community structures showed greater distances. Venn graph was used to analyze the common and unique OTU among different samples (groups) after the OTU results were obtained by clustering Higher numbers of shared OTU indicated greater similarity in community structure.

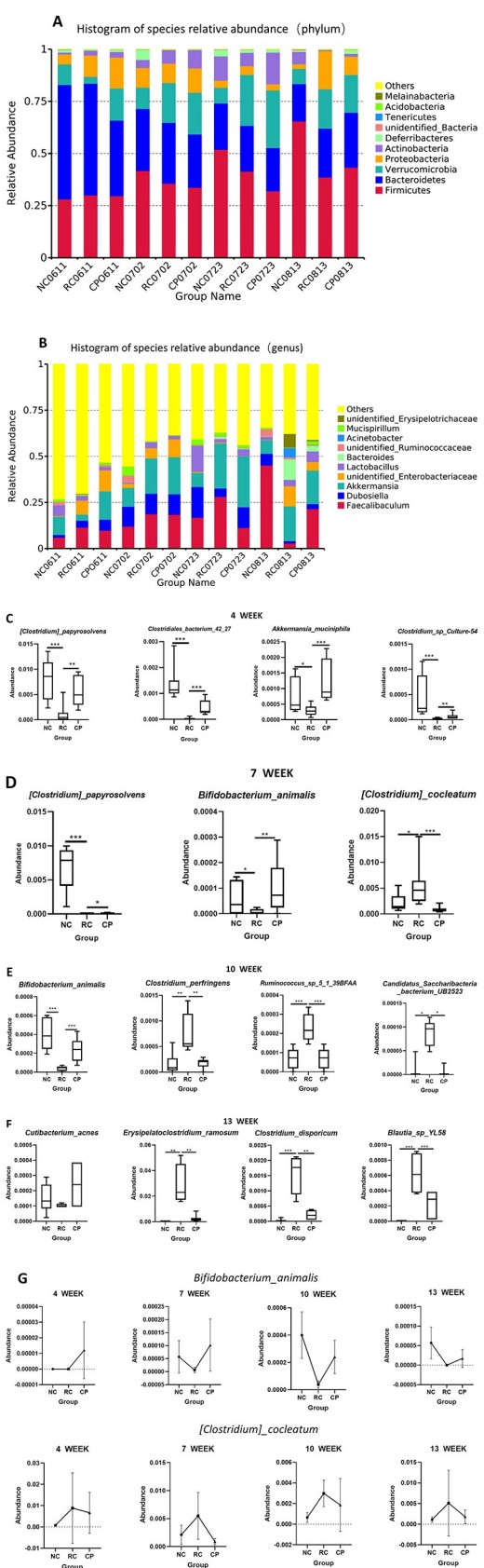

**Fig 4. Specific effects of the probiotic powder on the gut microbiota at different time points. (A, B)** Histogram of species relative abundance at the level of phylum and genus, respectively. Each color represents a species, and NC0611, NC0702, NC0723, and NC0813 represent the species changes of the NC group at the 4th, 7th, 10th, and 13th week, respectively, and the numbers for RC and CP indicate changes at these respective time points. **(C-F)** MetaStat diagrams of the different groups of mice at weeks 4, 7, 10, and 13, respectively. **(G)** Line diagrams of *Bifidobacterium animalis* and *Clostridium cocleatum*, respectively. One way analysis of variance (ANOVA), $^*p < 0.05$, $^{**}p < 0.01$, $^{***}p < 0.001$.

4, the probiotic powder promoted increased abundance of *Clostridium papyrosolvens*, *Clostridiales bacterium 42_27*, *Akkermansia muciniphila*, and *Clostridium sp. Culture-54*, which were reduced in CRC. At week 7, the probiotic powder promoted increased abundance of *Clostridium papyrosolvens* and *Bifidobacterium animalis*, which were reduced in CRC, and reduced the abundance of *Clostridium cocleatum* which is increased in CRC. At week 10, the probiotic powder promoted increased abundance of *Bifidobacterium animalis* in CRC, and reduced the abundance of *Clostridium perfringens*, *Ruminococcus sp. 5_1_39BFAA*, and *Candidatus Saccharibacteria bacterium UB2523*, which were increased in CRC. At week 13, the probiotic powder reduced the abundance of *Cutibacterium acnes* in CRC and inhibited that of *Erysipelatoclostridium ramosum*, *Clostridium disporicum*, and *Blautia sp. YL58*, which were increased in CRC. Re-analysis of the bacteria that showed statistically significant differences resulted in identification of two vital bacteria. The first was *Bifidobacterium animalis*, which was increased by the probiotic powder at all time points. The second was *Clostridium cocleatum*, which was reduced by the probiotic powder at all time points (Fig 4G). Our data suggested that these two bacteria were affected most by the probiotic powder in CRC.

## Probiotic powder inhibited CD4+ Foxp3+ Treg cells and promoted IFN-γ+ CD8+ T cells, CD4+ IL-4+ Th2 cells and CD19+ GL-7+ B cells

Various immune cells types and functions are affected by the gut microbiota, and changes to the gut microbiota can affect the host immune system, thus affecting CRC [14]. Treg cells are immunosuppressive cells that play an important role in tumor immunity [15]. Since our results showed that our probiotic powder inhibited colorectal cancer by regulating the gut microbiota, we evaluated whether the probiotic powder could affect Treg cells through regulation of the gut microbiota, resulting in anti-CRC effects. As shown in Fig 5A, the number of CD4+ Foxp3+ Treg cells in the RC group was significantly increased compared to that in the NC group. Treatment with the probiotic powder significantly reversed the CRC-induced increase in the number of CD4+ Foxp3+ Treg cells. These results suggested that the probiotic powder ameliorated colorectal cancer by inhibiting CD4+ Foxp3+ Treg cells.

As previously reported, the main function of Treg cells is to suppress effector T cells [16]. T cell function is controlled by TIGIT. The immune functions of T cells are negatively correlated with TIGIT expression [17]. CD8+ T cells are the main force of the anti-tumour immune response. As shown in Fig 5B, the number of IFN-γ+ CD8+ T cells was significantly reduced in the RC group compared with that in the NC group, intervention with the probiotic powder significantly reversed this decrease, suggesting that the probiotic powder promote the secretion of IFN-γ by CD8+ T cells to exert anti-tumour effects. In addition, We found that the number of CD4+ IL-4+ Th2 cells and the expression of TIGIT on the surface of CD4+ IL-4+ cells changed in response to CRC. As shown in Fig 5C, the number of CD4+ IL-4+ Th2 cells were significantly reduced in the RC group compared with that in the NC group. Intervention with the probiotic powder significantly reversed this decrease. (The changes of other effector T cells, including CD4+ CXCR3+ Th1 cells and CD4+ RoRrt+ Th17 cells in this study are not meaningful, as shown in Fig 5D). In addition, the probiotic powder significantly reversed CRC-induced increases in TIGIT expression on the surface of CD4+ IL-4+ Th2 cells. These

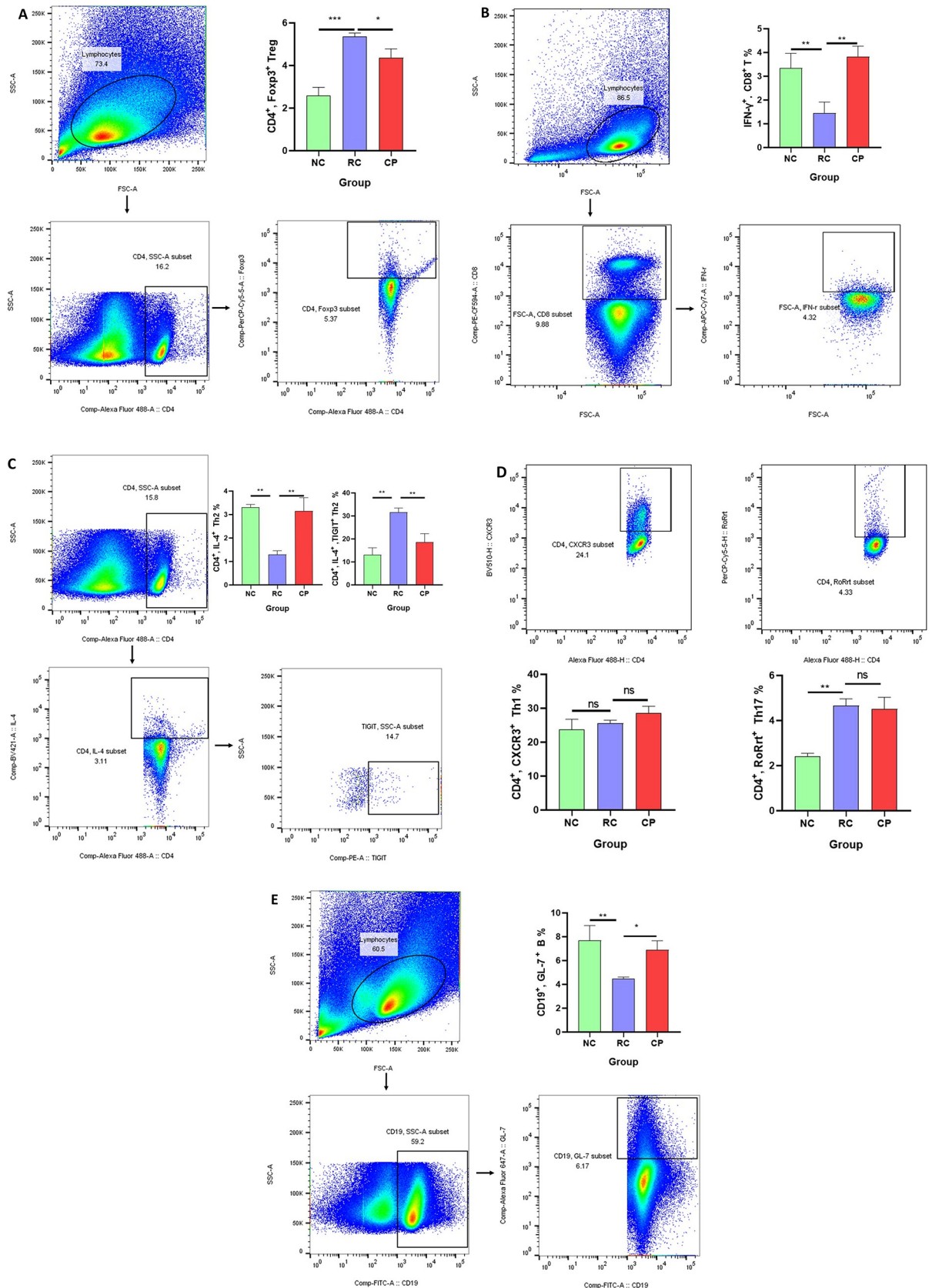

**Fig 5. Probiotic powder inhibited CD4⁺ Foxp3⁺ Treg cells, and promoted activation of IFN-γ⁺ CD8⁺ T cells, CD4⁺ IL-4⁺ Th2 cells and CD19⁺ GL-7⁺ B cells.** (A) Gating strategy for analysis of CD4⁺ Foxp3⁺ Treg cell subsets (gating within CD4 ⁺ T cells). (B) Frequency of IFN-γ⁺ CD8⁺ T cells (gating within CD8⁺T cells). (C) Frequency of CD4⁺ IL-4⁺ Th2 cells and expression of TIGIT on CD4⁺ IL-4⁺Th2 cells (gating within CD4 ⁺ T cells). (D) Frequency of CD4⁺ CXCR3⁺ Th1 cells and CD4⁺ RoRrt⁺ Th17 (gating within CD4 ⁺ T cells). (E) Frequency of CD19⁺ GL-7⁺ B cells (gating within CD19⁺ B cells). One way analysis of variance (ANOVA), *p < 0.05, **p < 0.01, ***p < 0.001.

results indicated that the probiotic powder inhibited Treg cells, resulting in increased number and activity of CD4⁺ IL-4⁺ Th2 cells.

Th2 cells can promote B cell maturation, proliferation, and antibody production to exert anti-tumor effects [18]. We evaluated the role of Th2 cell-mediated activation of B cells in CRC. As shown in Fig 5E, the number of CD19⁺ GL-7⁺ B cells in the RC group was significantly lower than that in the NC group. Intervention with the probiotic powder significantly increased the number of CD19⁺ GL-7⁺ cells compared to that in the RC group. These results suggested that the probiotic powder inhibited CD4⁺ Foxp3⁺ Treg cells, increased the number of IFN-γ⁺ CD8⁺ T cells, CD4⁺ IL-4⁺ Th2 cells and CD19⁺ GL-7⁺ B cells in the immune microenvironment of CRC.

## Probiotic powder increased the expression of pro-apoptotic protein

BAX and BCL-2 play important roles in regulating tumor growth. We measured the levels of BAX and Bcl-2 in tumor tissues to further characterize the mechanism of the gut microbiota-related anti-CRC effects of our probiotic powder [19]. As shown in Fig 6, the expression of BAX in the RC group was significantly lower than that in NC group. Treatment with the probiotic powder significantly reversed this decrease. In contrast, the probiotic powder did not affect the expression of Bcl-2.

## Discussion

Onset and progression of CRC is influenced by genetic and environmental factors with different etiological mechanisms [20]. Studies have shown that changes in the gut microbiome can initiate development of CRC, resulting in increased focus on management of CRC through modulation of the composition of the gut microbiota [21]. Probiotics have been shown to regulate the gut microbiota. Therefore, we developed a new probiotic powder to evaluate regulation of the microbiota to treat CRC. Probiotics are preparations of that contain viable, defined microorganisms in sufficient numbers [22] to alter the microflora in a compartment of the host, thereby exerting beneficial health effects [23]. The health benefits of prebiotics and probiotics in CRC have been demonstrated in animals and in humans [24]. In this study, we showed that a probiotic powder comprised of *Bifidobacterium* and *Lactobacillus* exerted anti-CRC effects. Furthermore, we characterized the mechanisms of these effects.

In this study, we found that probiotics could regulate the composition of the gut microbiome, promote beneficial bacteria, and inhibit harmful bacteria to inhibit CRC [25]. In CRC, several bacteria, such as *Bifidobacterium* and *Lactobacillus*, have shown anti-cancer properties in preclinical studies through several mechanisms, including inhibition of cell proliferation [26]. *Bifidobacterium animalis* has been shown to modulate the immune response, and to exert beneficial metabolic and anti-inflammatory effects [27]. However, few studies have evaluated the role of *Clostridium cocleatum* in CRC. We showed that our probiotic powder increased the abundance of *Bifidobacterium animalis* and reduced the abundance of *Clostridium cocleatum*. Future studies should evaluate the role of *Clostridium cocleatum* in CRC. In addition, future studies should evaluate the therapeutic effects of reducing the abundance of *Clostridium cocleatum*.

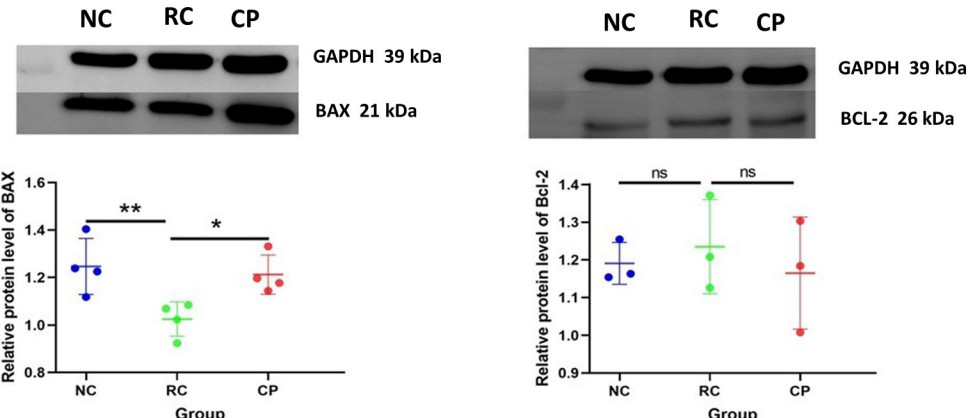

**Fig 6. Western blot of BAX and Bcl-2 in tumor tissue.** The expression of BAX (Molecular weight is 21 kDa), Bcl-2 (Molecular weight is 26 kDa), and GAPDH (Molecular weight is 39kDa) was measured using western blot. GAPDH was used as the loading control. Data are indicative of three separate experiments, and were analyzed using one way analysis of variance (ANOVA), $^*p < 0.05$, $^{**}p < 0.01$, $^{***}p < 0.001$.

Numerous studies have shown that the gut interacts closely with the host immune microenvironment [28], and plays an essential role in regulation of adaptive and innate immunity [29]. We evaluated whether regulation of the gut microbiota by probiotics resulted in changes in the immune microenvironment. Treg cells regulate homeostasis of the immune system and self-immune tolerance [30]. They can inhibit the anti-tumor effector of T cells and may alter the entire tumor immune environment [31]. CD8⁺T cells are an important part of tumor immunity, which clear tumors through a variety of mechanisms. IFN-γ secreted by CD8⁺T cells has anti-tumor activity and can control tumor growth. In addition, IFN- γ can enhance antigen presentation and inhibit tumor angiogenesis, which also plays an important role in cancer immune surveillance [32]. Th2 effector cells regulate B cell maturation, proliferation, and antibody production [33]. B cells regulate immune responses and inflammation through antibody production, and induce T cell activation and proliferation via antigen presentation. Our probiotic powder inhibited Treg cell activity, increased the abundance of IFN-γ⁺ CD8⁺ T cells, CD4⁺ IL-4⁺ Th2 cells and CD19⁺ GL-7⁺ B cells in the immune microenvironment of CRC.

T-cell immunoreceptor with immunoglobulin and tyrosine–based inhibitory motif domain is a member of the CD28 family and is mainly expressed on natural killer cells, CD4⁺, CD8⁺, and Treg cells [34]. It is crucial for balancing T cell activation and protection from autoimmunity [35]. T-cell immunoreceptor with immunoglobulin and tyrosine–based inhibitory motif domain expressed on the surface of T cells can bind to CD155 expressed by tumor cells, resulting in inhibition of the tumor killing effect of T cells. Our results showed that our probiotic powder reduced the expression of TIGIT on the surface of CD4⁺ IL-4⁺ Th2 cells. This result suggested that our probiotic powder increased the number of Th2 cells, and also increased the activity of Th2 cells. Reduced expression of TIGIT on Th2 cells promoted Th2 cell function to promote anti-tumor effects. Furthermore, reduced expression of TIGIT resulted in reduced binding to CD155 expressed by tumor cells, thus preventing immune escape.

Our study was subject to several limitations. First, the potential targets and specific mechanisms of the gut microbiota and immune cells requires further study. In addition, future studies should evaluate the effects of our probiotic powder in combination with other antineoplastic drugs such as Chinese herbal medicines or PD-L1 antibodies.

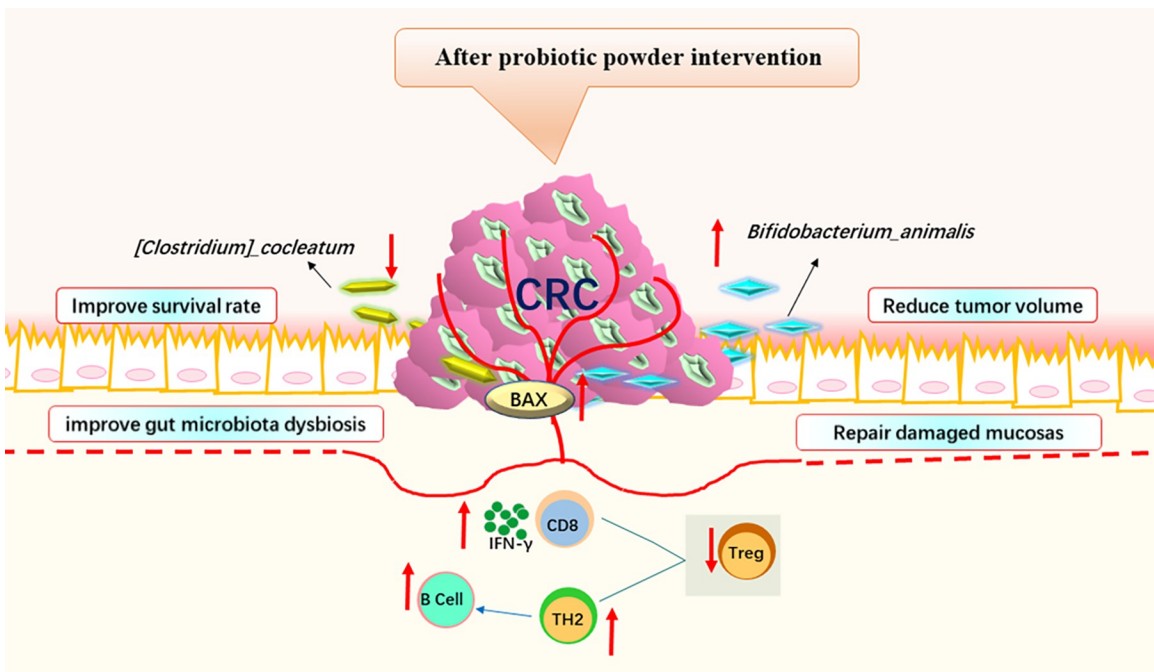

**Fig 7. Schematic diagram of the function of the probiotic powder in CRC.** The probiotic powder ameliorated CRC by regulating the gut microbiota (increased the abundance of *Bifidobacterium animalis* and reduced the abundance of *Clostridium cocleatum*), attenuating CD4[+] Foxp3[+] Treg cells, promoting the number of IFN-γ[+] CD8[+] T cells, increasing Th2 cell abundance and B cell abundance in the immune microenvironment of CRC, resulting in increased expression of BAX in CRC.

## Conclusion

In summary, we demonstrated that our probiotic powder ameliorated CRC by regulating *Bifidobacterium animalis*, *Clostridium cocleatum*, inhibiting Treg cells, promoting the number of IFN-γ[+] CD8[+] T cells, increasing Th2 cell abundance, inhibiting the expression of TIGIT in Th2 cells, and increasing B cell abundance in the immune microenvironment of CRC, resulting in increased expression of BAX in CRC. These results indicated that this probiotic powder has therapeutic potential for CRC (Fig 7).

## Supporting information

**S1 Data.**
(XLSX)

**S1 File. Images of the original western blots.**
(DOCX)

## Acknowledgments

We thank Dr. Richard Ho from Ningxia Medical University for text correction and Dr. Meng Chen from Tianjin Novogene Biotech Co., Ltd for data analysis and helpful discussion. Thanks to Charlesworth Author Services for scientific editing and proof-reading.

## Author Contributions

**Data curation:** Qian Cao.

**Formal analysis:** Miao Liu.

**Funding acquisition:** Xiangguo Duan.

**Methodology:** Yuhan Xia, Jinhua Tian.

**Supervision:** Bin Ma, Xiangguo Duan.

**Visualization:** Xiangguo Duan.

**Writing – original draft:** Xiaojuan Yang.

**Writing – review & editing:** Jian Chen, Chunxia Su, Xiangguo Duan.

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
