## [Decision Letter · Decision Letter 0]

7 Jun 2022

PONE-D-22-11955Probiotic powder ameliorates colorectal cancer by regulating Bifidobacterium animalis and Clostridium cocleatum, with CD4+ FOXP3+, CD4+ IL-4+ T cells, CD19+ GL-7+ B cellsPLOS ONE

Dear Dr. Duan

Thank you for submitting your manuscript to PLOS ONE. After careful consideration, we feel that it has merit but does not fully meet PLOS ONE’s publication criteria as it currently stands. Therefore, we invite you to submit a revised version of the manuscript that addresses the points raised during the review process.

While the reviewers feel that the study is potentially interesting, they have critical concerns about the writing style of the manuscript, quality of Figures, experimental design, and interpretation of the results. Moreover, the authors showed beneficial effects of probiotic powders on many target cells, however, the authors do not integrate these findings in the manuscript.

We look forward to receiving your revised manuscript.

Kind regards,

Hiroyasu Nakano, M.D., Ph.D.

Academic Editor

PLOS ONE

Journal Requirements:

Reviewers' comments:

Reviewer's Responses to Questions

**Comments to the Author**

1. Is the manuscript technically sound, and do the data support the conclusions?

Reviewer #1: Partly

Reviewer #2: No

2. Has the statistical analysis been performed appropriately and rigorously? 

Reviewer #1: I Don't Know

Reviewer #2: No

3. Have the authors made all data underlying the findings in their manuscript fully available?

Reviewer #1: No

Reviewer #2: Yes

4. Is the manuscript presented in an intelligible fashion and written in standard English?

Reviewer #1: No

Reviewer #2: No

5. Review Comments to the Author

Reviewer #1: In the manuscript entitled "Probiotic powder ameliorates colorectal cancer by regulating Bifidobacterium animalis and Clostridium cocleatum, with CD4+Foxp3+, CD4+IL-4+ T cells, CD19+GL-7+ T cells" by Yang et al, the authors assess the effect of their probiotic power on AOM/DSS-mediated colorectal cancer model in mice. They show that administration of the probiotic powder suppresses tissue damages, a decrease of goblet cells, and focal necrosis in the cancer model (Fig. 1A-C). The probiotic powder also improves survival (Fig. 1D) and reduces tumor volume (Fig. 1E). The authors reveal that the probiotic powder diminishes the cancer model-associated alteration of microbiota (Fig. 2 and Fig. 3). They also show that the probiotic powder suppresses the change of lymphocytes in the cancer model: an increase in the number of Treg cells, a decrease in Th2 cells, upregulation of TIGHT expression on the surface of Th2 cells, and a reduction of GL7+ B cells (Fig. 4). Finally, the authors show the probiotic powder suppresses the reduction of the pro-apoptotic protein BAX in the cancer model.

While the study shows multiple beneficial effects of the probiotic powder on a mouse cancer model, I feel that appropriate integration of such diverged findings is necessary. The authors should explain the relationships among cancer, microbiota, lymphocytes, and BAX expression. In addition, the manuscript seems to be thoroughly improved possibly with the aid of experienced scientists, because no references are cited in Methods and Results sections, figure legends are poor, and the resolution of figures are too low.

Specific comments:

The title of the manuscript should be refined.

In Results section, the authors should first explain the purpose of each experiment, citing appropriate references.

In Fig. 1A-C, the authors should indicate tissue damages, reduction of goblet cells, and necroptosis using different types of markers, such as arrows and arrow heads.

In Fig. 4, it is unknown why the authors analyzed only limited types of lymphocytes. The authors should widely analyze many types of lymphocytes, and then specify the affected populations by the cancer model and probiotic powder. Especially, CD8+ T cells seems to be important.

In Fig. 5, which types of cells diminish the expression of BAX after induction of cancer model?

Minor comments

In Figure 5, the molecular weight of protein is generally indicated as kDa rather than KB.

Reviewer #2: In this study, Yang and her colleagues demonstrated that probiotic powder prepared in their laboratory improves the survival rate of AOM-DSS-induced colon adenoma in mice. They have also found that their compound favored the reduction of Foxp3+CD4 T cells and the expansion of IL-4+CD4 T cells and GL7+ GC B cells in the spleen. The shift in the composition of T and B cells may be attributed to the changes in the colon microbiota, that is; the increase of Bifidobacterium and reduction of Clostridia species. These data are altogether interesting and may contribute to the treatment of human disease of colorectal cancer.

However, this reviewer raises several concerns that should be addressed, as described below, before consideration for publication to PLOS ONE.

Major comments

1. As far as this reviewer understands, the incidence of adenoma is tightly associated with gut inflammation in AOM-DSS model. The primary output of this model is the number and/or volume of adenoma and not the long-term survival. However, according to Figure 1A,B,C,and E, the difference of adenoma volume between RC and CP was marginal at best. Perform a statistical analysis, provide the p-value, and number of animals used in Figure 1, please. At this point, it appears that the conclusion that probiotic powder ameliorates colorectal cancer is not supported by scientific evidence.

2. Related to the Major comment 2, please show the body weight change of 3 groups of animals after the administration of AOM-DSS. It should be examined whether the probiotic powder used by authors reduced or enhanced the degree of inflammation induced by DSS.

3. It is not clear whether the probiotic powder used in this study was a mixture generally used worldwide or the one optimized by themselves. It should be explained more carefully how they tuned the composition of probiotic used in this study.

Minor comments

1. Provide the molecular weight of DSS used in this study, please.

2. Some external grammatical editing must be performed.

6. PLOS authors have the option to publish the peer review history of their article (what does this mean?). If published, this will include your full peer review and any attached files.

Reviewer #1: No

Reviewer #2: No

---

## [Author Response · Author response to Decision Letter 0]

4 Aug 2022

Dear editor and reviewer：

I am very happy to receive your decision. First of all, thank you for your interest and affirmation to my research. It is my honor to submit my manuscript to your Journal and I am very cherishing the opportunity to be accepted by your Journal. Therefore, I have made serious amendments to the questions you raised, and the specific answers will be displayed in the following reply form. I am very appreciated for your consideration and looking forward to your reply, hope everything goes well with your work and everything goes your way. As follows are the specific responses.

Editor

Thank you for giving us the opportunity to submit a revised draft for publication in the journal of PLOS ONE. The following are the changes we made based on your suggestion:

1. manuscript style requirements

We have modified the style of the article in accordance with the requirements of PLOS ONE, including figure style, table style, references and so on.

2. Information regarding the experiments involving animals

In the content of the “Mice” in the Methods part, we supplement the animal experiment and make sure that it includes the description of methods of sacrifice, efforts to alleviate suffering and so on.

3. Data Availability statement

In the data availability statement, we have specified where the minimal data set underlying the results described in our manuscript can be found, and we promise that all the minimal data in our articles will be fully available. The link in the Data Availability statement (https://www.ncbi.nlm.nih.gov/bioproject/PRJNA669487) is the Illumina sequencing raw data we uploaded to NCBI. In addition, we have uploaded a supporting Information files for study’s minimal underlying data. In addition, we have submitted all other results in the manuscript in the form of a supporting file, which is called “Minimal data set”

4. Uncropped and unadjusted images underlying all blot

We uploaded the original image without uncropped and unadjusted images. The file name is "Images of the original western blots"

5. Ethics statement

We have deleted the ethics statement in other parts of the article and only described the ethics statement in the methods section

Reviewer #1

Thank you for your affirmation to my work and your valuable suggestions, which has greatly improved the quality and accuracy of our manuscript. The following are the changes we made based on your suggestion:

1. Explain the relationships among cancer, microbiota, lymphocytes, and 

BAX expression

We think this is an excellent suggestion, and we made changes to this suggestion, mainly in the description of the results, as well as the conclusion. We think Probiotic powder can ameliorate CRC by regulating gut microbiota (increased the abundance of Bifidobacterium animalis and reduced the abundance of Clostridium cocleatum). The changes of gut microbiota in turn regulate the immune environment of CRC mice, including attenuating CD4+ Foxp3+ Treg cells, promoting CD4+ IL-4+Th2 cells and inhibiting the expression of TIGIT in CD4+ IL-4+Th2 cells, promoting CD19+ GL-7+B cells. The above changes of immune cells induced effective anti-tumor immune response in CRC mice, which enhanced the ability of immune cells to kill tumor cells. As a result, the biological function of tumor cells has been changed, resulting in the high expression of pro-apoptotic proteins and then apoptosis. The number of apoptotic tumor cells increased, resulting in the inhibition of CRC.

2. The title of the manuscript 

Thanks for your suggestions, we have refined the title, and now the title is “Probiotic powder ameliorates colorectal cancer by regulating Bifidobacterium animalis, Clostridium cocleatum, and immune cell composition ”

3. About references are cited in Methods and Results sections, figure

legends and the resolution of figures

Thanks for your suggestions. First of all, we make some changes to the description of the methods and results, and add references, such as Ref. 10-19. Secondly, we modified the figure legends and make a full description of the figure legends by referring to other manuscript. Finally, we adjusted the figures and improved the resolution of the figures.

4. In Results section, the authors should first explain the purpose of each

experiment, citing appropriate references.

Thank you for pointing this out, we have added the purpose of each experiment in the results section and inserted relevant references, such as references 13-19.

5. The issue in Fig. 1A-C

Your suggestion really means a lot to us. We have added different colors of arrows to indicate different tissue damages, such as the red arrow represents the normal goblet cells, the green arrow represents the crypt abscess, the blue arrow represents inflammatory necrosis, and the black arrow represents inflammatory cell infiltration. 

6. The issue in Fig. 4

Thank you for your positive comments. In order to explore the effect of probiotic powder on immunity of mice by regulating gut microbiota, we are not limited to the analysis of these specific cells. In addition to these cells, we also analyzed CD11C+MHCII+DC cells, CD4+,CXCR3+Th1 cells and CD4+,CCR6+Th17 cells. But in these cells, only CD4+ Foxp3+ Treg cells, CD4+ IL-4+ Th2 cells and CD19+ GL-7+ B cells changed significantly. Therefore, we believe that Probiotic powder play a role mainly by affecting these three immune cells in CRC mouse model.

7. The issue in Fig. 5

Thank you for pointing this out, we have made a supplementary description in the results and figure legends, the extracted protein is a protein from mouse colorectal cancer tissue, and then the expression of BAX in this tissue was detected. In addition, we fell sorry for our carelessness, in our resubmitted manuscript, the molecular weight of protein has been indicated as kDa. 

Reviewer #2

We fell great thanks for your professional review work on our article. As you are concerned, there are several problems that need to be addressed. According to your nice suggestions, we have made extensive corrections to our provious draft, the detalied corrections are listted below.

1. The issue in Fig 1

Thank you for pointing this out, after careful consideration, we adjusted Fig 1E. from the comparison of the maximum tumor volume between RC and CP to the average tumor volume comparison between RC and CP，through the analysis, there was significant difference between the two groups (P < 0.05). The average tumor volume of different mice and the number of mice in each group were shown in Table2.

2. Whether the probiotic powder reduced or enhanced the degree of

inflammation induced by DSS.

We sincerely appreciate you valuable communts. In determining the effect of probiotic powder on inflammation caused by DSS, we mainly judged by HE staining, HE shows that probiotic powder can significantly reduced the degree of inflammation of CRC. We are sorry that we did not consider the weight change to reflect the degree of inflammation at the same time. Thank you very much for your suggestion. We will include this factor in later studies.

3. About probiotic powder

Thank you for pointing this out, the probiotic powder used in this study is our own optimized mixture, mainly based on the previous experimental results, the use standards of related additives, and the guidance of professional instructors (Professor Fan Yanli of Food discipline in Ningxia University, Professor Yang Jianhong, doctoral supervisor of Pharmacy in Ningxia Medical University, and Xia Yuhan, director of the Department of Nutrition in General Hospital of Ningxia Medical University).

4. Provide the molecular weight of DSS used in this study.

Thanks for your suggestions, we have added the molecular weight of DSS and the molecular weight of AOM in the Methods. As you can see, Azoxymethane(CAS No.:25843-45-2, molecular weight :74.08) and dextran sulfate sodium salt(CAS No.:9011-18-1, molecular weight : ~40,000) .

5. Grammatical editing problem.

As soon as we received the decision, we submitted the manuscript to Charlesworth Author Services （CAS）for correct English language usage, grammar, punctuation and spellings. We have submitted a version of the manuscript edited by CAS (file name is “Revised Manuscript by CWS”) and an Editorial certificate (file name “CWS_Editorial_Certificate”)

---

## [Decision Letter · Decision Letter 1]

9 Sep 2022

PONE-D-22-11955R1Probiotic powder ameliorates colorectal cancer by regulating Bifidobacterium animalis, Clostridium cocleatum, and immune cell compositionPLOS ONE

Dear Dr. Duan

Thank you for submitting your manuscript to PLOS ONE. After careful consideration, we feel that it has merit but does not fully meet PLOS ONE’s publication criteria as it currently stands. Therefore, we invite you to submit a revised version of the manuscript that addresses the points raised during the review process. As one of the reviewers mentioned, the authors need to include a point-by-point reply to the reviewers' comments, which clearly shows what points the authors have changed in the revised manuscript.

We look forward to receiving your revised manuscript.

Kind regards,

Hiroyasu Nakano, M.D., Ph.D.

Academic Editor

PLOS ONE

Journal Requirements:

Reviewers' comments:

Reviewer's Responses to Questions

**Comments to the Author**

1. If the authors have adequately addressed your comments raised in a previous round of review and you feel that this manuscript is now acceptable for publication, you may indicate that here to bypass the “Comments to the Author” section, enter your conflict of interest statement in the “Confidential to Editor” section, and submit your "Accept" recommendation.

Reviewer #1: (No Response)

Reviewer #2: All comments have been addressed

2. Is the manuscript technically sound, and do the data support the conclusions?

Reviewer #1: Yes

Reviewer #2: Yes

3. Has the statistical analysis been performed appropriately and rigorously? 

Reviewer #1: Yes

Reviewer #2: I Don't Know

4. Have the authors made all data underlying the findings in their manuscript fully available?

Reviewer #1: Yes

Reviewer #2: Yes

5. Is the manuscript presented in an intelligible fashion and written in standard English?

Reviewer #1: Yes

Reviewer #2: Yes

6. Review Comments to the Author

Reviewer #1: The response to reviewers was not good. The authors should cite the original comments by reviewers for point-by-point responses. In fact, some comments are overlooked.

In Fig. 4, the authors need to analyze Th1 cells, Th17 cells, and CD8+ T cells.

Many letters in Fig. 2A, Fig. 2B, Fig. 3B, Fig. 4A, Fig. 4B, Fig. 4C are not visible because of their inappropriate size and poor resolution.

Reviewer #2: Authors have adequately responded to this reviewer's requests. The manuscript would be ready for publication to the PLOS ONE.

7. PLOS authors have the option to publish the peer review history of their article (what does this mean?). If published, this will include your full peer review and any attached files.

Reviewer #1: No

Reviewer #2: No

---

## [Author Response · Author response to Decision Letter 1]

12 Sep 2022

Dear editor and reviewer：

Thank you for giving us the opportunity to resubmit a revised draft of the manuscript “Probiotic powder ameliorates colorectal cancer by regulating Bifidobacterium animalis, Clostridium cocleatum, and immune cell composition” for publication in the Journal of PLOS ONE. We appreciate the time and effort that the editor and reviewers dedicated to providing insightful comments and valuable suggestions to our paper. I am very appreciated for your consideration and looking forward to your reply, hope everything goes well with your work and everything goes your way. As follows are the specific responses.

Editor

Thank you for giving us the opportunity to submit a revised draft for publication in the journal of PLOS ONE. The following are the changes we made based on your suggestion:

1. Please review your reference list to ensure that it is complete and 

correct.

Our reply:

Thank you for pointing this out, we checked our reference list, the 23rd reference was incomplete and we re-cited it to make sure the references were complete and correct. In addition, our references do not contain articles that have been retracted, and all references are searchable on Pubmed.

Reviewer #1

We feel great thanks for your professional review work on our articles. As you are concerned, there are several problems that need to be addressed. Based on your nice suggestions, we have made extensive corrections to our previous manuscript, the detalied corrections are listted below.

1. In Fig. 4, the authors need to analyze Th1 cells, Th17 cells, and CD8+ 

T cells.

Our reply:

Thank you for pointing this out, this suggestion really means a lot to us. In our experimental design, we wanted to investigate the mechanism of the role of key gut microbiota in colorectal cancer by affecting Treg cells. Since Treg cell mainly plays an immunosuppressive function in cancer, we examined the changes of various types of effector T cells (CD4+ T cells), including Th1 cells, Th2 cells and Th17 cells, while CD8+ T cells were not considered. Ultimately, Th2 cells were found to play a major function. We apologize for not including other effector T cells with no statistically significant changes in Figure 4. Thanks to your valuable suggestion, we have added Th1 cells and Th17cells to Figure 4 (Fig. 4C), and described accordingly in the results and methods. As shown below:

2. Many letters in Fig. 2A, Fig. 2B, Fig. 3B, Fig. 4A, Fig. 4B, Fig. 4C 

are not visible because of their inappropriate size and poor resolution.

Our reply:

Thanks for your suggestions, we have resized and sharpened all the figures in the article including Fig. 2A, Fig. 2B, Fig. 3B, Fig. 4A, Fig. 4B and Fig. 4C. The resolution and sharpness of the figures have been improved.

Thanks again for your reconsideration of publishing our manuscript in your journal. We are looking forward to hearing from you soon!

With best wishes

Sincerely

---

## [Decision Letter · Decision Letter 2]

2 Oct 2022

PONE-D-22-11955R2Probiotic powder ameliorates colorectal cancer by regulating Bifidobacterium animalis,Clostridium cocleatum, and immune cell compositionPLOS ONE

Dear Dr. Duan,

Thank you for submitting your manuscript to PLOS ONE. After careful consideration, we feel that it has merit but does not fully meet PLOS ONE’s publication criteria as it currently stands. Therefore, we invite you to submit a revised version of the manuscript that addresses the points raised during the review process. One of the reviewers feels that the authors have not responded to the reviewer's comments. Please sincerely respond to the comments, otherwise we do not consider the revision of the manuscript.

We look forward to receiving your revised manuscript.

Kind regards,

Hiroyasu Nakano, M.D., Ph.D.

Academic Editor

PLOS ONE

Reviewers' comments:

Reviewer's Responses to Questions

**Comments to the Author**

1. If the authors have adequately addressed your comments raised in a previous round of review and you feel that this manuscript is now acceptable for publication, you may indicate that here to bypass the “Comments to the Author” section, enter your conflict of interest statement in the “Confidential to Editor” section, and submit your "Accept" recommendation.

Reviewer #1: (No Response)

2. Is the manuscript technically sound, and do the data support the conclusions?

Reviewer #1: Partly

3. Has the statistical analysis been performed appropriately and rigorously? 

Reviewer #1: I Don't Know

4. Have the authors made all data underlying the findings in their manuscript fully available?

Reviewer #1: No

5. Is the manuscript presented in an intelligible fashion and written in standard English?

Reviewer #1: No

6. Review Comments to the Author

Reviewer #1: 1. The authors should analyze CD8+ T cells, because the study deals with cancer.

2. Most letters are still not visible in Figures.

7. PLOS authors have the option to publish the peer review history of their article (what does this mean?). If published, this will include your full peer review and any attached files.

Reviewer #1: No

---

## [Author Response · Author response to Decision Letter 2]

12 Oct 2022

Dear reviewer：

Thank you for your letter and for the reviewers’ comments. We really appreciate the reviewers for your valuable comments and we have studied comments carefully and revised the manuscript. We hope the revised manuscript could be acceptable for publication in the Journal of PLOS ONE. The responds to the reviewer’s comments are as flowing, thank you in advance for your time and effort!

1.The first comments of Reviewer :The authors should analyze CD8+ T cells, because the study deals with cancer. 

Our reply:

Thank you for pointing this out, we have analyzed CD8+ T cells, including the changes of IFN-γ+ CD8+ T cells and we added the results to Fig 4B, and described accordingly in the results , discussion, conclusion and methods. As shown below.

CD8+ T cells are the main force of the anti-tumour immune response, which clear tumors through a variety of mechanisms. IFN-γ secreted by CD8+T cells has anti-tumor activity and can control tumor growth. In addition, IFN- γ can enhance antigen presentation and inhibit tumor angiogenesis, which also plays an important role in cancer immune surveillance. Therefore, in this study, as the Reviewer said, it is very important to analyze CD8+ T cells.

As shown in Fig 4B, the number of IFN-γ+ CD8+ T cells was significantly reduced in the RC group compared with that in the NC group, intervention with the probiotic powder significantly reversed this decrease, suggesting that the probiotic powder promote the secretion of IFN-γ by CD8+ T cells to exert anti-tumour effects.

Based on the results of our analysis, our conclusions are as follows: In summary, we demonstrated that our probiotic powder ameliorated CRC by regulating Bifidobacterium animalis, Clostridium cocleatum, inhibiting Treg cells, promoting the number of IFN-γ+ CD8+ T cells, increasing Th2 cell abundance, inhibiting the expression of TIGIT in Th2 cells, and increasing B cell abundance in the immune microenvironment of CRC, resulting in increased expression of BAX in CRC. These results indicated that this probiotic powder has therapeutic potential for CRC. 

2.The second comments of Reviewer :Most letters are still not visible in Figures.

Our reply:

In order to make the figures clearer, we have carefully revised the figures again. The resolution of the modified figures is 300dpi, and the dimensions, file size of the figures have been revised in strict accordance with the requirements of of PLOS ONE. In addition, we have uploaded our figure files to the PACE to ensure that our figures meet PLOS requirements.

 We tried our best to revised the manuscript, we appreciate for Editors and Reviewers’ warm work earnestly, and hope that the correction will meet with approval. Once again thank you very much for you.

Best regards

Sincerely yours

---

## [Decision Letter · Decision Letter 3]

18 Oct 2022

PONE-D-22-11955R3Probiotic powder ameliorates colorectal cancer by regulating Bifidobacterium animalis,Clostridium cocleatum, and immune cell compositionPLOS ONE

Dear Dr. Duan,

Thank you for submitting your manuscript to PLOS ONE. After careful consideration, we feel that it has merit but does not fully meet PLOS ONE’s publication criteria as it currently stands. Therefore, we invite you to submit a revised version of the manuscript that addresses the points raised during the review process. Although the authors have responded to most comments by Reviewer 1, Reviewer 1 still feels that the size of letters are too small to read them in Figures. Thus, the authors should increase the size of letters in the revised Figures.

Please include the following items when submitting your revised manuscript:A rebuttal letter that responds to each point raised by the academic editor and reviewer(s). You should upload this letter as a separate file labeled 'Response to Reviewers'.A marked-up copy of your manuscript that highlights changes made to the original version. You should upload this as a separate file labeled 'Revised Manuscript with Track Changes'.An unmarked version of your revised paper without tracked changes. You should upload this as a separate file labeled 'Manuscript'.If applicable, we recommend that you deposit your laboratory protocols in protocols.io to enhance the reproducibility of your results. Protocols.io assigns your protocol its own identifier (DOI) so that it can be cited independently in the future. For instructions see: https://journals.plos.org/plosone/s/submission-guidelines#loc-laboratory-protocols. Additionally, PLOS ONE offers an option for publishing peer-reviewed Lab Protocol articles, which describe protocols hosted on protocols.io. Read more information on sharing protocols at https://plos.org/protocols?utm_medium=editorial-email&utm_source=authorletters&utm_campaign=protocols.

We look forward to receiving your revised manuscript.

Kind regards,

Hiroyasu Nakano, M.D., Ph.D.

Academic Editor

PLOS ONE

Journal Requirements:

Reviewers' comments:

Reviewer's Responses to Questions

**Comments to the Author**

1. If the authors have adequately addressed your comments raised in a previous round of review and you feel that this manuscript is now acceptable for publication, you may indicate that here to bypass the “Comments to the Author” section, enter your conflict of interest statement in the “Confidential to Editor” section, and submit your "Accept" recommendation.

Reviewer #1: (No Response)

2. Is the manuscript technically sound, and do the data support the conclusions?

Reviewer #1: Partly

3. Has the statistical analysis been performed appropriately and rigorously? 

Reviewer #1: I Don't Know

4. Have the authors made all data underlying the findings in their manuscript fully available?

Reviewer #1: Yes

5. Is the manuscript presented in an intelligible fashion and written in standard English?

Reviewer #1: Yes

6. Review Comments to the Author

Reviewer #1: Because I still do not see many letters in Figures, I would like to allow the editor or editorial office to make a decision.

7. PLOS authors have the option to publish the peer review history of their article (what does this mean?). If published, this will include your full peer review and any attached files.

Reviewer #1: No

---

## [Author Response · Author response to Decision Letter 3]

19 Oct 2022

Dear reviewer：

Thank you for your letter and for the reviewers’ comments. We really appreciate the reviewers for your valuable comments and we have followed comments carefully and revised the manuscript. We hope the revised manuscript could be acceptable for publication in the Journal of PLOS ONE. The responds to editor and the reviewer’s comments are as flowing, thank you in advance for your time and efforts!

Reviewer #1:Because I still do not see many letters in Figures, I would like to allow the editor or editorial office to make a decision.

Our reply:

We fell great thanks for the reviewer’s professional review work on our article. As the reviewer’s concerned, there still some letters are too small in Figures, we made adjustment again to our Figures. The letters in revised Figures are now of a higher quality and the resolution, dimensions, file size of the figures have been revised in strict accordance with the requirements of PLOS ONE. In addition, we have uploaded our figure files to the PACE to ensure that our Figures meet PLOS requirements. The following are the specific changes:

1.The letters in Fig 2A , Fig 2B, 3C are fully enlarged. Take Fig 2A as an example, the revised Figure is as follows:

2. AS Fig 3A and Fig 3B have too much contents, resulting in blurred letters. We make appropriate adjustments to Fig 3A and Fig 3B, and divided Fig 3A into two parts, named Fig 3A and Fig 3B respectively. The contents of Fig 3B is divided into four parts, which are named as Fig 3C, Fig 3D, Fig 3E and Fig 3F respectively. The adjusted Figures are clearer and the letters are visible. Take Fig 3A as an example:

3. The letters of Fig 4A , Fig 4B, Fig 4C , Fig 4D and Fig 4E are not clear due to the improper combination of the Figures. Therefore, we adjusted the way in which these Figures are combined. Take Fig 4E as an example, the adjusted image is as follows:

The Figures modification made the letters much clearer, the contents of the Figures has not changed, and does not affect the structure and description of our manuscript.

 We tried our best to revised the manuscript, we appreciate to Editors and Reviewers’ hard work, and hope that the corrections could meet you requirement. Once again thank you very much for your nice patients.

Best regards

Sincerely yours

---

## [Decision Letter · Decision Letter 4]

21 Oct 2022

Probiotic powder ameliorates colorectal cancer by regulating Bifidobacterium animalis,Clostridium cocleatum, and immune cell composition

PONE-D-22-11955R4

Dear Dr. Duan,

We’re pleased to inform you that your manuscript has been judged scientifically suitable for publication and will be formally accepted for publication once it meets all outstanding technical requirements.

Kind regards,

Hiroyasu Nakano, M.D., Ph.D.

Academic Editor

PLOS ONE

Additional Editor Comments (optional):

Reviewers' comments:

Reviewer's Responses to Questions

**Comments to the Author**

1. If the authors have adequately addressed your comments raised in a previous round of review and you feel that this manuscript is now acceptable for publication, you may indicate that here to bypass the “Comments to the Author” section, enter your conflict of interest statement in the “Confidential to Editor” section, and submit your "Accept" recommendation.

Reviewer #1: (No Response)

2. Is the manuscript technically sound, and do the data support the conclusions?

Reviewer #1: (No Response)

3. Has the statistical analysis been performed appropriately and rigorously? 

Reviewer #1: (No Response)

4. Have the authors made all data underlying the findings in their manuscript fully available?

Reviewer #1: (No Response)

5. Is the manuscript presented in an intelligible fashion and written in standard English?

Reviewer #1: (No Response)

6. Review Comments to the Author

Reviewer #1: (No Response)

7. PLOS authors have the option to publish the peer review history of their article (what does this mean?). If published, this will include your full peer review and any attached files.

Reviewer #1: No

---

## [Editor Report · Acceptance letter]

13 Dec 2022

PONE-D-22-11955R4 

Probiotic powder ameliorates colorectal cancer by regulating *Bifidobacterium animalis, Clostridium cocleatum*, and immune cell composition 

Dear Dr. Duan:

I'm pleased to inform you that your manuscript has been deemed suitable for publication in PLOS ONE. Congratulations! Your manuscript is now with our production department. 

Kind regards, 

on behalf of

Professor Hiroyasu Nakano 

Academic Editor

PLOS ONE